# Among Gerontogens, Heavy Metals Are a Class of Their Own: A Review of the Evidence for Cellular Senescence

**DOI:** 10.3390/brainsci13030500

**Published:** 2023-03-16

**Authors:** Samuel T. Vielee, John P. Wise

**Affiliations:** 1Pediatrics Research Institute, Department of Pediatrics, University of Louisville, Louisville, KY 40202, USA; 2Department of Pharmacology and Toxicology, University of Louisville, Louisville, KY 40202, USA

**Keywords:** aging, gerontogens, heavy metals, cellular senescence, neurodegeneration

## Abstract

Advancements in modern medicine have improved the quality of life across the globe and increased the average lifespan of our population by multiple decades. Current estimates predict by 2030, 12% of the global population will reach a geriatric age and live another 3–4 decades. This swelling geriatric population will place critical stress on healthcare infrastructures due to accompanying increases in age-related diseases and comorbidities. While much research focused on long-lived individuals seeks to answer questions regarding how to age healthier, there is a deficit in research investigating what aspects of our lives accelerate or exacerbate aging. In particular, heavy metals are recognized as a significant threat to human health with links to a plethora of age-related diseases, and have widespread human exposures from occupational, medical, or environmental settings. We believe heavy metals ought to be classified as a class of gerontogens (i.e., chemicals that accelerate biological aging in cells and tissues). Gerontogens may be best studied through their effects on the “Hallmarks of Aging”, nine physiological hallmarks demonstrated to occur in aged cells, tissues, and bodies. Evidence suggests that cellular senescence—a permanent growth arrest in cells—is one of the most pertinent hallmarks of aging and is a useful indicator of aging in tissues. Here, we discuss the roles of heavy metals in brain aging. We briefly discuss brain aging in general, then expand upon observations for heavy metals contributing to age-related neurodegenerative disorders. We particularly emphasize the roles and observations of cellular senescence in neurodegenerative diseases. Finally, we discuss the observations for heavy metals inducing cellular senescence. The glaring lack of knowledge about gerontogens and gerontogenic mechanisms necessitates greater research in the field, especially in the context of the global aging crisis.

## 1. Introduction: Two Faces to a Toxic Aging Crisis

We are experiencing an unprecedented rise in the age of our global population due to increased lifespan; a phenomenon described as an aging crisis. Estimates suggest by 2030, 12% of the world’s population will be geriatric, with this estimate reaching 16% in 2050 and 23% in 2100 [1]. This aging crisis will impact developed nations to a greater extent, where the geriatric population will reach 20% by 2030. The term “geriatric” describes an individual who has at least a 2% chance of mortality within a year, though geriatrics are colloquially defined to be individuals 65 years of age or older [2]. Such a dramatic increase in the geriatric population will place significant constraints on our healthcare systems worldwide, increasing the need for medical care, personnel, and devices to treat aging-related health conditions.

Compounding this aging issue, we are faced with global environmental pollution of innumerable chemicals that threaten our health—a separate crisis that challenges aging health in two key ways, which we refer to as a “toxic aging coin” [3]. On the heads side of this toxic aging coin, we consider how life stage (e.g., juvenile vs. middle-aged vs. geriatric) impacts the toxic effects of chemicals. On the tails side, we consider how chemicals accelerate aging, or how they act as gerontogens.

We know chemical exposures early in life can have long-lasting effects, which contribute to health status in later life, as described by the Developmental Origins of Health and Disease (DOHaD) hypothesis [4]. We recognize elderly populations are a vulnerable group in regard to health issues and threats to public health (e.g., epidemics), but we rarely recognize geriatrics as a distinct demographic for toxicological consideration. Previously, geriatrics were often dismissed in toxicology as already having deteriorating health or too close to death to consider for risk assessments. However, with increased lifespan potentially contributing 3–4 more decades of geriatric life, there is a critical need to better understand the toxicology of aging and improve risk assessment for all ages. This need is affirmed by the NIEHS in Theme 1 of their strategic plan, “Advancing Environmental Health Sciences encompasses the study of all levels of biological organization… at all stages across the lifespan, from preconception through old age” [5].

We are only just starting to understand how some environmental pollutants act as gerontogens, i.e., chemicals that induce or accelerate biological aging processes. At a clinical level, gerontogens may contribute to a broad spectrum of age-related diseases and comorbidities, affecting all organ systems, e.g., osteoporosis, cancers, diabetes, and dementias. We recognize mortality and age-related diseases are linked to or exacerbated by environmental pollutants; thus, it is imperative we better understand gerontogenic mechanisms [6,7]. To better understand gerontogens, it is essential to recognize two contexts of aging: chronological age (i.e., age according to birth year) and biological aging (i.e., cellular/tissue aging caused by gerontogenic factors). The presentation of biological aging may have traits specific to each type of cell, tissue, or organ; yet, some aging effects are consistently observed and are collectively referred to as the “Hallmarks of Aging” [8]. These hallmarks are nine distinct changes in biological processes that increase with normal aging: (1) stem cell exhaustion; (2) altered intercellular communication; (3) genomic instability; (4) telomere attrition; (5) epigenetic alterations; (6) loss of proteostasis; (7) dysregulation of nutrient sensing; (8) mitochondrial dysfunction; and (9) cellular senescence [8]. Consideration for the hallmarks of aging is further described for specific organs, such as the “Hallmarks of Brain Aging” [9]. In addition to the nine hallmarks of aging, brain aging includes (10) dysregulation of neuronal calcium homeostasis; (11) oxidative stress; (12) impaired molecular waste disposal; (13) impaired DNA repair capacity; (14) glial cell activation; and (15) impaired adaptive stress response signaling [9]. As such, these aging hallmarks become incredibly useful tools for assessing the gerontogenic effects of environmental pollutants.

Perhaps the most pertinent and readily measured aging hallmark is cellular senescence, a permanent state of cellular growth arrest with a distinct shift in metabolism and secretory phenotype. Cellular senescence represents a permanent shift in a cell’s phenotype to an aged phenotype, likely caused by mechanisms involving one or multiple other aging hallmarks. Phenotypically, senescent cells exhibit increased expression of p16^INK4a^ and p21 following the activation of either the p53-p21 or Rb-p16^INK4a^ pathways, respectively [10,11]. In addition to an increased expression of p16^INK4a^ and p21, some of the most identifiable characteristics of senescent cells include the expression of senescence-associated β-Galactosidase (SA-β-Gal), mitochondrial dysfunction, increased lysosomal content, and a large, irregularly shaped cell body [12]. The accumulation of senescent cells can promote further senescence in neighboring cells through secreted factors or senescence-associated secretory phenotype (SASP) factors [13]. Importantly, senescent cells accumulate with age in all tissue types, and their selective clearance with senolytic drugs can attenuate aging-related pathology and clinical symptoms [14,15,16,17]. Hence, improving our understanding of gerontogenic mechanisms leading to cellular senescence will be imperative to improve healthy aging and diminishing adverse aging effects of chemicals.

Here, we will discuss the evidence to consider heavy metals as a class of gerontogens. Our discussion will start with identifying why heavy metals are a health concern. We will review unique heavy metal exposures to geriatric populations. Then, we will discuss what an aged brain looks like, and how heavy metals induce some brain aging effects in human and animal studies. Finally, we will review and discuss what is known about how heavy metals induce cellular senescence.

## 2. Materials and Methods

### 2.1. Search Strategy

Literature searches utilized PubMed and Google Scholar for literature relevant to heavy metals, cellular senescence, hallmarks of aging, and physiological processes of an aging brain. Searches began in October 2022 and concluded in January 2023. Searches were specific to metals and cellular senescence (e.g., “arsenic cellular senescence”) or metals and relevant biomarkers for cellular senescence (e.g., “arsenic p21 expression”). Additional searches included metals and the broad term hallmarks of aging (e.g., “arsenic hallmarks of aging”) or a metal and a specific hallmark of aging (e.g., “arsenic genomic instability”). To ensure all relevant research was found, search terms often included the abbreviation for the metal and the term “heavy metal” alongside the previously described searches (e.g., “Pb lead heavy metal cellular senescence”). This was conducted specifically in searches relevant to lead to avoid confusion with its homonyms and limit the presence of irrelevant research. Searches for current information regarding heavy metal contamination and human exposure followed a similar pattern, using the search term “heavy metals environmental contamination.” Searches for information regarding the physiology of the aging brain used terminology specific to brain regions or structures alongside the term aging and the physiological process (e.g., gray matter attrition aging, blood–brain barrier leakage aging). Searches for specific neurodegenerative diseases relevant to these diseases often followed a similar query with regards to up-to-date statistics (e.g., Alzheimer’s disease prevalence), the pathology of the disease (e.g., Alzheimer’s disease pathology), or the significance of cellular senescence or the biomarkers of senescence in these diseases (e.g., Alzheimer’s disease cellular senescence, Alzheimer’s disease p21 expression). Articles cited within relevant publications were also selected and examined.

### 2.2. Exclusion Strategy

Articles were excluded from searches if the model utilized was not a cell culture model, an animal model, or a study of human participants. This was performed to ensure the information discussed is wholly relevant to human health. Articles were also prioritized based on the date of publication to ensure that the information gathered and the data discussed are not outdated. Particular emphasis on publication date was given to research discussing environmental contamination and prevalence of neurodegenerative diseases.

## 3. Environmental Heavy Metal Pollution Poses a Significant Health Threat

“Heavy metals” refers to metals and metalloids studied in the context of environmental pollution, toxicology, and adverse effects on biological life [18]. They are part of our natural environment and may be released at toxic levels in events, such as volcanic eruptions or forest fires, but much of the concern for heavy metal pollution stems from human industry. Heavy metals pervade our societies and have innumerable valuable uses ranging from electronics, pigments and dyes, and metals to agricultural practices, and many of these industries release contaminated effluents into the environment that can lead to dangerous human exposures [19,20]. Heavy metals pollute our air, water, soil, and food supplies, and are one of the most dangerous forms of environmental pollution, as they generally cannot be metabolized or degraded into less toxic forms, and they can persist in our environment for decades [19,21,22,23,24,25]. Importantly, among the top ten chemicals on the 2022 ATSDR Hazard Priority List, the top three are heavy metals: arsenic, lead, and mercury [26]. Their neurotoxic effects are widely recognized across the age spectrum [27,28].

## 4. Unique Heavy Metal Exposures in Geriatric Populations

In addition to environmental pollution, geriatric populations experience some chemical exposures unique to their age group. Polypharmacy is a prominent concern in geriatric individuals contending with multiple comorbidities and deteriorating health. Though it is out of the scope of this review, studies show geriatric polypharmacy has adverse effects such as cognitive impairment, frailty, malnutrition, and mortality [29,30]. Polypharmacy presents challenges to geriatric health involving heavy metals in two key ways: (1) heavy metals are a common impurity in polymeric excipients (i.e., inert chemicals/substances required for pharmaceutical production), including lead, arsenic, mercury, chromium, nickel, cadmium, and iron, and (2) can significantly alter toxicokinetic responses to heavy metals [31].

The use of medical devices increases with age and presents unique chemical exposures, particularly metal-on-metal (MoM) joint replacements. One study estimated hip and knee replacements will have an increased incidence of 84% and 183%, respectively, from 2001 to 2026 in New Zealand [32]. MoM hip replacements fail at a greater rate than prostheses made from other materials, with the 6-year failure rate for MoM hip replacements reaching up to 50% [33]. Failed MoM joint replacements can result in metal debris released inside the body, resulting in unusually high levels of cobalt (Co) and chromium (Cr) [34]. Such internal exposures have localized (e.g., inflammation, necrosis, and osteolysis) and systemic effects (e.g., neurological effects, visual impairment, cardiomyopathy, and hypothyroidism) [34,35]. Patients have exhibited toxic levels of Co and Cr as high as 669.4 nmol/L and 664 nmol/L [36]. Metallosis caused by failed MoM joint replacements induces various toxic effects, including a host of neurological deficits (e.g., peripheral neuropathy, sensory loss, tinnitus, and gray matter atrophy) [37,38,39]. While there is no estimate for a global trend of the prevalence of MoM prosthetics, there is a clear need for an assessment of the toxic outcomes geriatric individuals experience from exposure to a wide array of medical devices.

## 5. Considerations for Brain Aging

To understand how heavy metals might induce brain aging, we will first review how we assess an aged brain. Several distinct behavioral changes occur in aged populations, including decreased locomotor activity, decreased cognitive abilities, impaired learning and memory abilities, and depression [40]. Many of these behaviors can be readily assessed in human or animal studies. These aging-related behaviors are accompanied by morphological and biochemical changes in the brain, such as decreased white matter volume, gray matter volume, and neurotransmitter levels [41,42]. The aged brain is often described with neurodegeneration, neuroinflammation, loss of dendritic densities, changes in synaptic plasticity, and changes in neurotransmission [43,44]. Aging-related loss of gray matter is apparent in the cerebral cortex, caudate nucleus, pallidum, amygdala, and hippocampus, while loss of white matter is observed in the anterior thalamic radiations, anterior limb of the internal capsule, and cerebellum [45]. These changes are also accompanied by decreased cortical thickness, hippocampal volume, and frontal lobe volume [46]. Aged brains exhibit an accumulation of a wide variety of senescent cells, including neurons, astrocytes, microglia, endothelial cells, and oligodendrocytes, each of which affects brain health distinctly [47,48,49]. Cellular senescence in the brain results in decreased synaptic plasticity and contributes to the overall behavioral changes associated with an aging brain. These effects can be reversed or alleviated with the application of senolytic drugs that selectively induce apoptosis in senescent cells, demonstrating the pertinence of cellular senescence to brain aging [48].

### 5.1. Heavy Metals and Cellular Senescence Observations in Neurodegenerative Diseases

Most often researchers consider how heavy metals contribute to age-related neurodegenerative diseases such as Alzheimer’s disease, Parkinson’s disease, and amyotrophic lateral sclerosis. Many studies have reported links between heavy metal exposure and these neurodegenerative diseases [27,50,51,52]. More recently, cellular senescence has gained attention in these neurodegenerative diseases [53,54]. 

Alzheimer’s disease (AD), a progressive neurodegenerative disease, is the most prevalent form of dementia and is often considered a primary phenotype of brain aging. In 2022, the global number of people with AD (i.e., AD with dementia, prodromal AD, or preclinical AD) was estimated to affect over 416 million people worldwide and accounted for 22% of all people aged 50 and above [55]. AD is largely a sporadic disease, with less than 10% of cases attributed to genetic causes [56]. Behavioral changes related to AD include progressively deteriorating cognitive abilities and are most often diagnosed by a mental exam or cognitive assessment [57]. Pathologically, AD is characterized by cortical, hippocampal, and amygdala atrophy, preceded by the accumulation of amyloid beta plaques and neurofibrillary tangles [58]. Several heavy metals are linked to AD pathogenesis and progression, including iron, copper, manganese, zinc, lead, aluminum, mercury, and cadmium [59,60]. Recent evidence demonstrated increased populations of senescent neurons, astrocytes, and microglia early in AD progression [55]. Further, the brain tissue of patients who suffered from early onset AD (ages 35–50) had significantly increased expression of p16^INK4a^ in astrocytes, with a greater burden of increased p16^INK4a^ expression in the prefrontal cortex, an area known to atrophy during AD progression [58,61]. Cell culture studies show amyloid-β_1–42_ exposure induces senescence in adult neural stem cells [62]. Finally, a recent human trial reported improved AD symptoms with senolytic therapy [16]. Altogether, there is ample evidence for cellular senescence playing a pathological role in AD.

Parkinson’s disease (PD) was estimated to affect over 8.5 million people in 2019 with increasing global trends of PD burden over the previous two decades [63]. Similar to AD, PD is an idiopathic disease with approximately 10% of cases caused by genetics [64]. Heavy metals linked with PD etiology include iron, zinc, copper, lead, mercury, manganese, aluminum, arsenic, cadmium, and selenium [65,66]. PD is attributed to the death of dopaminergic neurons in the substantia nigra and loss of dopaminergic axon terminals in the striatum, which presents as motor symptoms of PD [67,68]. PD can also be classified with or without dementia, depending on if cognitive abilities are affected by the disease [69]. PD patients exhibited increased SA-β-Gal expression in cerebrospinal fluid [54]. Astrocytes cultured with paraquat, a pesticide associated with PD, exhibited increased expression of SA-β-Gal, p16^INK4a^ mRNA, and mRNA of SASP-related cytokines (IL-1α, IL-6, and IL-8) [70,71]. PD patients also presented with other hallmarks of aging, such as loss of proteostasis, mitochondrial dysfunction, and altered cell-to-cell communication [72].

Amyotrophic lateral sclerosis (ALS) has a worldwide prevalence of 4.4 in 100,000 and has steadily increased over the last 50 years [73]. ALS has an average age of onset of 55 years with a short life expectancy (3–5 years) after diagnosis [74,75]. ALS is a rapidly progressive neurodegenerative disease characterized by the degeneration of motor neurons in the brainstem and spinal cord [74]. Epidemiological evidence points to exposure to heavy metals contributing to ALS risk, particularly lead, mercury, cadmium, and selenium [76,77]. Mouse models suggest that ALS progression is driven by cellular senescence, as disease progression was correlated with the expression of cellular senescence markers (p16^INK4a^, p21, and SA-β-Gal) in microglia, astrocytes, and spinal motor neurons [78]. The post mortem brain tissue of individuals who suffered from ALS had a greater proportion of cells expressing cellular senescence markers such as p16^INK4a^ in astrocytes and p21 in both astrocytes and neurons [79]. 

Further details regarding exposure to heavy metals or metalloids and the development of neurodegenerative diseases can be found in Table 1. It should be noted that two metals (iron and zinc) thoroughly discussed in PD and AD were left out of this table; these metals serve physiological roles within cells/tissues that contribute to disease process, while human exposures to these metals are not known to contribute to disease. For reviews on the roles of iron and zinc in AD and PD, see reviews [80,81,82,83,84].

### 5.2. Age-Related BBB/NVU Impairment

The most significant changes to occur in brain aging are decreased function and impaired integrity of the blood–brain barrier (BBB) and the neurovascular unit (NVU) [117,118]. The NVU is a structure that encompasses the brain’s vasculature, composed of neurons, endothelial cells, astrocytic end feet, pericytes, and microglia, which create a physical barrier between the blood and brain parenchyma [119]. Endothelial cells and pericytes work together to secrete the basal lamina that surrounds blood vessels and to create the BBB. The BBB is a physical and metabolic barrier that creates a tightly sealed vascular wall, preventing exogenous materials from entering the brain [120]. This BBB formation makes the vascular endothelial cells of the brain phenotypically distinct from those elsewhere in the body, expressing specific tight junction proteins to form the BBB. These tight junction proteins include occludins, claudins, junctional adhesion molecules, and cytosolic proteins (e.g., zona-occludins-1, -2, and -3) [121]. The NVU and BBB gradually deteriorate with age and some brain regions exhibit greater leakage than others, such as the hippocampus (specifically CA1 and the dentate gyrus) [122,123,124,125]. Further, the olfactory bulb and circumventricular organs lack any BBB protection, including the pineal gland, neurohypophysis, area postrema, subfornical organ, median eminence, and the vascular organ of the lamina terminalis [126]. Increased permeability of the NVU will also lead to increased susceptibility to the neurotoxic effects of various chemicals, such as heavy metals.

There are many reports of heavy metals affecting BBB integrity and function. A thorough discussion of heavy metal toxicity at the BBB is beyond the scope of this paper, see references for more detailed reviews [127,128,129]. Lead (Pb) is the most often reported heavy metal for BBB toxicity and shows a higher affinity for accumulating in brain endothelial cells than other brain cell types [130]. Pb can also cross the BBB using Ca ATPase pumps and induce BBB permeability [131,132,133,134,135]. Other heavy metals observed to induce BBB permeability include cadmium, copper, mercury, and arsenic [136,137,138,139,140,141,142]. Many studies using these heavy metals have expanded their focus to assess changes in BBB proteins expressed in the vasculature, reporting decreased tight junction proteins and increased adherens proteins as indicators of BBB impairment. Decreased occludin is the tight junction protein most often reported decreased in heavy metal BBB toxicity [132,133,143,144,145]. Several studies also report metal-induced decreases in claudins and zona occludins proteins [133,142,144,145,146,147]. Finally, several reports observed elevated expression of adherens proteins (e.g., ICAM-1 and VCAM-1) in endothelial cells following heavy metal exposure, indicating a shift in endothelial cells toward an inflammatory phenotype [148,149,150]. Such an increase in adherens proteins increases interaction with circulating white blood cells and promotes their intravasation into the brain parenchyma [151,152,153].

Cellular senescence may contribute to aging-related BBB leakage and has gained a lot of attention in recent years [154,155,156,157]. Cellular senescence within the NVU or BBB is observed in the brains of patients with Alzheimer’s and Parkinson’s diseases [71,158,159,160]. Several studies with aged mice or aging-accelerated mice demonstrated increased BBB leakiness associated with senescent endothelial cells [161,162,163]. Much of the mechanisms for brain endothelial cell senescence contributing to BBB leakage remain to be elucidated. A seminal paper using an in vitro BBB model and BubR1 hylomorphic mice reported senescent endothelial cells accumulated in aged cerebrovasculature and senescent endothelial cells exhibited impaired BBB integrity with reduced tight junction coverage [161].

## 6. Heavy Metals Induce Cellular Senescence

Here, we present the evidence for heavy metals as a class of gerontogen. The literature for heavy-metal-induced cellular senescence specific to the brain is limited, so we broaden our focus here to heavy-metal-induced senescence in any mammalian tissue or cell type. A review of the research for heavy metals contributing to all the hallmarks of aging is beyond the scope of this review, so we limit our discussion to metals inducing cellular senescence. References to metals inducing other aging hallmarks are provided where applicable. Chromium, arsenic, cadmium, lead, copper, uranium, and silver have all been observed to induce cellular senescence in a variety of cell types and research models, whereas, zinc, nickel, and copper have limited evidence, highlighting the need for additional research. A summary of senescence markers that increased in response to heavy metal or metalloid exposure is provided in Table 2.

### 6.1. Arsenic

Arsenic (As) is one of the most important environmental contaminants when considering metals, given the widespread global exposure to this metal through sources such as drinking water. Doses as low as 0.25 μM As induce a nearly two-fold expression of p16 and p21 in skin cell culture models [164]. In animal models, 50 ppm As increased p16 expression even after As exposure had been terminated [165]. A human study in West Bengal, India, found peripheral blood mononuclear cells of individuals exposed to contaminated drinking water exhibited increased protein levels of p16, p21, p53, SA-β-Gal, and SASP-associated inflammatory cytokines [166]. Other studies found increased expression of SASP and SA-β-Gal in both animal and cell culture models [167,168,169]. One study observed genomic instability, telomere dysfunction, and a senescent phenotype in glioma cell culture after 48 h incubation with 4 or 8 μM As_2_O_3_ [170]. Altogether, evidence for As-induced cellular senescence is found in cell culture, animal, and human studies. 

### 6.2. Lead

Lead pollution in air and drinking water has a long history of challenging human health. Pb exposure in animal models increased G1 cell cycle arrest in hematopoietic stem cells and increased the expression of SA-β-Gal in CD90^+^CD45^-^ cells following low-dose exposure to Pb [171]. Studies utilizing rat models observed increased gene expression of SASP inflammatory cytokines (IL-1α, IL-1β, IL-6, and IL-8) in pups whose mothers had been treated with 0, 2000, or 4000 ppm lead acetate during gestation [172]. This study also reported senescence-associated changes in cells of the subgranular zone, with decreased expression of *Pcna* and *Apex1* mRNA levels (correlated with cell proliferation), increased expression of *Chek1* and *Cdknc1c* mRNA levels (cell cycle arrest marker), and increased expression of γH_2_AX protein levels (DNA damage markers). These effects indicate that Pb-induced genotoxicity contributed to senescent phenotypes in the hippocampus. Animal and cell culture models demonstrated the gerontogenic effects of Pb with evidence of premature senescence in the subgranular zone in one animal study, emphasizing the regional or localized effects of senescence.

### 6.3. Cadmium

Cadmium (Cd) induces senescent phenotypes and several other hallmarks of aging. Cell culture models demonstrated the activation of the p53-p21 pathway following treatment with 15 μM cadmium chloride (CdCl_2_) [173]. Primary neurons isolated from fetal mice exhibited increased expression of p53, p21, and p16 following 24 h exposure to 10 μM Cd [174]. This study reported an increased expression of SASP and Senescence-Associated-β-Galactosidase (SA-β-Gal). One study using zebrafish exposed to 24 μM CdCl_2_ reported increased expression of lipid peroxidation (MDA measured in zebrafish adult hepatic tissue) and increased SA-β-Gal expression in primary human dermal fibroblasts [197]. Luo et al. (2021) reported increased markers of genomic instability and cellular senescence in mesenchymal stromal cells derived from the bone marrow of Sprague–Dawley rats [175]. In summary, there is sufficient evidence of Cd-induced senescence in animal and cell culture models.

### 6.4. Chromium

Chromium (Cr), in the valence state of hexavalent chromium (Cr(VI)), readily induced cellular senescence following the activation of the p53-p21 pathway [176,177,178]. Culturing BEAS-2B lung fibroblasts with 0.5 or 1 μM Cr(VI) induced the expression of SA-β-Gal [179]. In hepatocyte cell culture, 10 nM Cr(VI) induced premature senescence with increased expression of SA-β-Gal and SASP proteins (COTL1, ENO1, and PRDX2) [180]. Hepatocytes treated with 10 nM Cr(VI) also show increased mRNA expression of inflammatory cytokines (IL-6 and IL-8) and senescence markers (p53 and p21) [181]. In the context of other hallmarks of aging, hepatocytes treated with 10 nM Cr(VI) altered mitochondrial morphology while inducing cellular senescence (confirmed by SA-β-Gal expression) [182]. A human study demonstrated occupational exposure to Cr(VI) increased blood serum levels of Apolipoprotein J/Clusterin, another aging biomarker [198]. Cr(VI) is well described to induce senescence in cell culture and aging effects in human populations.

### 6.5. Copper

Elevated intracellular copper (Cu) is a phenotype of senescent cells [199]. Cell culture models utilizing glioblastoma cells found Cu increased the expression of senescence-associated mRNA (*ApoJ*, *p21*, *p16*, and *TGF-β*) and proteins (p21 and p16) [183]. These changes were also observed in human lung fibroblasts with senescent morphologies [184]. Another study reported HeLa cells treated with Cu increased the expression of p53 and p21 while promoting oxidative stress and mitochondrial dysfunction [185]. This study reported Cu increased G2/M arrest, which is associated with cellular senescence [185]. Additional evidence demonstrated oxidative stress from Cu exposure induced premature senescence mediated by p38 in fetal lung fibroblasts [186]. In human lung fibroblasts, Cu induced SA-β-Gal expression and altered proteostasis [187]. Interestingly enough, treatment of Cu with the drug resveratrol decreased the expression of p16^INK4a^ in the brains of aged mice, as well as several other hallmarks of brain aging, suggesting a non-monotonic response to Cu in brain aging [200]. While there is strong evidence of Cu inducing senescence in cell culture, a potential anti-aging property of Cu highlights the knowledge gaps that exist for metals toxicity, cellular senescence, and aging.

### 6.6. Uranium

Depleted uranium (DU) is the form of processed uranium remaining after uranium enrichment, maintaining roughly 60% of the radiation found in uranium ores [201]. This form of uranium is often used in military operations, medical devices, and the manufacturing of aircraft and boats, contributing to both occupational and environmental exposures [201]. DU maintains the same genotoxic effects of its base element and is well known to cross the BBB to induce neurotoxicity [202]. Drinking water exposure in animal studies has shown DU increased expression in the inflammatory genes characteristic of SASP [188]. Another animal study showed 150 mg/L DU exposure in drinking water induced lipid peroxidation in the brain and behavioral changes in 7-month-old rats after only 2 weeks of exposure [203]. There is evidence DU induces both cytotoxic and clastogenic effects [204]. Unprocessed forms of U also induced genomic instability and drove cellular senescence via ionizing radiation [205,206]. Thus, U and DU induce senescence in animal models and should be given special consideration due to their unique exposures.

### 6.7. Iron

Iron has a much more complex role in cellular senescence. The literature on iron inducing cellular senescence is sparse and is likely not a major concern. One paper reported iron induced cellular senescence associated with fibrotic diseases in mice, with a significant increase in p21- and p16^INK4A^-positive cells six days after iron delivery [196]. Another paper induced cellular senescence in microglia by overloading cell cultures with iron, resulting in an SASP phenotype [207]. The vast majority of literature pertaining to iron and cellular senescence focuses on the distinct accumulation of iron and iron dyshomeostasis in senescent cells [208,209,210]. As such, many studies have begun targeting these iron features for senolytic drugs, particularly inducing ferroptosis [211,212,213,214]. For more detailed reviews on the roles of iron dyshomeostasis and accumulation in senescent cells, see reviews [210,215,216,217].

### 6.8. Heavy Metals with Less Evidence for Cellular Senescence

While the aforementioned metals demonstrate a clear induction of cellular senescence or alterations to cellular physiology consistent with the hallmarks of aging, metals such as nickel, cobalt, silver, and zinc present limited evidence of these characteristics and necessitate greater research efforts.

Nickel (Ni) is often utilized to mimic hypoxic environments and induce oxidative stress to activate HIF-1α, a hypoxia-inducible transcription factor [189]. This increase in HIF-1α induces growth arrest and p21 expression in various cell lines, a strong indication of cellular senescence [218]. Additionally, Ni depolarizes the mitochondrial membrane potential and leads to mitochondrial dysfunction in HeLa cells [219].

Cobalt (Co) also acts as a hypoxia mimic and increases the expression of HIF-1α [220]. In cardiac myocytes, Co-induced HIF-1α expression increased the expression of p21, SASP, SA-β-Gal, and ROS [190]. An additional study modeled age differences in cell culture by using cells with a low or high population doubling as a substitute for chronological age. This study reported cells at both “ages” exhibited genomic instability following Co exposure [191]. Co induced SA-β-Gal expression in “older” cells, but “younger” cells exhibited decreased p38 and caspase-10 gene expression [191].

Silver (Ag) may be the most unique metal to consider as a gerontogen due to its increased use as silver nanoparticles (AgNPs) [221]. AgNPs induced oxidative stress and increased inflammatory cytokines (IL-1β, MCP-1, and MIP-2) in vitro after 24 h exposure [192]. Other studies showed AgNPs promoted a SASP phenotype and induced SA-β-Gal expression, genomic instability, and cell cycle arrest in the G2/M phase [193,194,222].

Limited evidence indicates zinc (Zn) mediated cellular senescence by inducing the overexpression of Nox1 and p21 in vascular smooth muscle cells [195]. Salazar et al. (2017) also indicated Zn accumulated within the mitochondria of these cells and promoted ROS production [195].

## 7. Concluding Remarks

Concerns for the health effects of heavy metals have risen over the years as public outrage concerning the environmental pollution of metals increases in intensity. Heavy metals are known to contaminate soil and water used for drinking and agricultural irrigation [21,22,23]. Additionally, heavy metal contamination is pervasive in some food sources such as seafood, crops, and, most strikingly, in baby food products [24,25,223]. Collectively, this poses a significant public health threat as heavy metals cannot be degraded or metabolized into less toxic forms and can bioaccumulate in various tissues.

A rise in aging comorbidities will likely coincide with the projected increase in human lifespan, threatening global healthcare infrastructure. While much aging research is focused on healthy aging, research into adverse factors that accelerate or exacerbate aging lags behind. Heavy metals are of particular concern, given their widespread distribution and links to many age-related diseases. Cellular senescence is a key outcome of gerontogen exposure, which can be readily assessed with simple screening assays.

This knowledge gap for gerontogenic mechanisms of heavy metals persists despite clear evidence indicating their ability to induce a variety of aging hallmarks in cell culture, animal, and human models. The localized aging effects of gerontogens demand a more subtle assessment of aging phenotypes and the comorbidities that develop from affected tissues. The threat of brain gerontogens is further compounded by a lack of knowledge concerning cellular senescence in the NVU. While there is evidence regarding senescent phenotypes in the cell types comprising the NVU, we lack an understanding of how the interplay between these components influences one’s exposure to heavy metals and gerontogens.

There is a significant need to address our limited understanding of toxic outcomes in geriatric individuals. Geriatrics experience unique toxic outcomes resulting from polypharmacy and medical devices and likely respond differently to toxicants due to their degenerating biology [224]. This is in stark contrast to a juvenile or middle-aged individual with a developing or stable biology, respectively. Despite this difference in biological contexts, there is little to no special consideration given to the toxic outcomes experienced by geriatrics.

The increasing average age of the global population will no doubt result in a concomitant increase in age-related diseases, painting a bleak picture for our impending future. To better prepare for this inevitable shift in population structure, we need significantly more research into gerontogenic mechanisms and the unique toxic outcomes in geriatric individuals.

## Figures and Tables

**Table 1 brainsci-13-00500-t001:** Associations between heavy metals/metalloids and neurodegenerative diseases.

Metal or Metalloid	Demographics of Human Study	Animal Model (Species/Strain)	Exposure	Associated Neurodegenerative Disease(s)	Observations	Mechanistic Details	Refs.
Arsenic	133 men and 301 women older than 40 years, living in Cochran or Parmer County, Texas	N/A	2.19–15.26 μg/L for 1–80 years	AD	Decreased language scores, visuospatial skills, and executive functioning.	As exposure induces ROS generation and DNA damage, leading to broad neurodegeneration [85,86]. Dimethyl arsenic increases expression of Aβ following APP upregulation, suggesting a mechanism related to AD [87].	[88]
	Prevalence of AD and other dementias compared to average As concentration in topsoil from eleven European countries	N/A	7–18 ppm	AD	Diagnosis and prevalence within country. Dementia Composite Index (product of AD prevalence and mortality rank within group) plotted against As soil concentration.		[89]
	Prevalence of PD patients in Taiwan compared to As soil contamination	N/A	2.18–11.47 mg/kg in soil	PD	Diagnosis of PD correlated with As soil concentration using Local Moran’s *I* and spatial regression analysis.		[90]
	6 ALS patients (3 men, 3 women)	N/A	N.R.	ALS	Serum As concentration had a strong positive correlation with disease duration.		[91]
Lead	121 PD patients and 414 controls	N/A	Determined via participant interview	PD	PD diagnosis and regression, indicating a two-fold increase in PD diagnosis following Pb exposure.	Pb crosses the BBB by substituting for divalent metals [92]. Pb enters neurons and astroglia, inducing mitochondrial damage and initiating C-caspase cascade activation [92]. This disrupts metabolism, alters neurotransmission, and induces neurodegeneration [92]. Pb is linked to AD as it is shown to increase Aβ accumulation and reduce Aβ clearance [87].	[93]
	17 patients with a definitive ALS diagnosis aged 47–85 years old from Oslo, Norway; 10 controls were aged 26–77 years old	N/A	N.R.	ALS	ICP-MS indicates a significantly greater CSF concentration of Pb in ALS patients compared to controls.		[94]
	V/A	*Macaca fascicularis*	1.5 mg/kg lead acetate during infancy (until day 400), assessed at age 23	AD	Increased *APP* expression, elevated Aβ levels; decreased DNA methyltransferase and higher oxidative DNA damage		[95]
	N/A	Long–Evans hooded rats, 24 h old	200 ppm lead acetate for 20 days or 18–20 months	AD	Upregulation of *APP* and increased expression of Aβ.		[96]
	N/A	Kunming Mice	Prenatal exposure to 0.1%, 0.5%, or 1% lead acetate, lasting until weaning	AD	Altered performance on Morris water maze. Increased immunoreactivity in cerebral cortex. Increased expression of Aβ.		[97]
	N/A	Long–Evans hooded rats	50 ppm lead acetate in drinking water for 90 days	PD	Loss of basal dopamine, 3,4-dihydroxyphenylace tic acid, and homovanillic acid levels.		[98]
Cadmium	64-year-old man	N/A	Acute (60 min) inhalation	PD	3 months after exposure patient exhibited with bradykinesia, stooped posture, cogwheel muscle rigidity, short-stepped gait, and disturbed freezing and righting reflexes.	Cd is associated with ALS pathogenesis by reducing Cu/Zn SOD enzymes, disrupting BBB integrity, and inducing glutamate toxicity [99]. Cd induced oxidative stress and inflammation in rat cortex and hippocampus, and increased synthesis of Aβ, resembling AD pathology [87].	[100]
	Fixed brain tissue from 8 ALS, 4 PDC, and 5 control subjects who lived in Guam.	N/A	N.R.	ALS	Gray and white matter from ALS patients had elevated concentrations of Cd in all regions except for the basal ganglia.		[101]
	4064 participants from the 1999–2004 NHANES aged greater than 60 years with available blood Cd data. Participants included 51 individuals who suffered AD mortality.	N/A	N.R.	AD	Greater blood Cd concentration is strongly associated with increased risk of AD mortality.		[102]
	17 patients with a definitive ALS diagnosis aged 47–85 years old from Oslo, Norway; 10 controls were aged 26–77 years old	N/A	N.R.	ALS	ICP-MS indicates a significantly greater CSF concentration of Cd in ALS patients compared to controls.		[94]
Chromium	6 ALS patients (3 men, 3 women)	N/A	N.R.	ALS	Whole blood Cr was higher in ALS patients when compared to reference values of Italian population. Whole blood Cr concentration is also correlated with disease duration in this study.	No exact mechanism has been elucidated for Cr(VI) neurotoxicity. Lipid peroxidation, apoptosis, altered acetylcholinesterase, and inflammation observed in Cr(VI) neurotoxicity across multiple taxa [39].	[91]
	AD patients (12 men, 41 women) aged 68–94 years old, controls (100 men, 117 women) were patients at the same hospital being treated for optical disorders	N/A	N.R.	AD	Multivariable logistic regression and odds ratios indicate an association between high blood concentrations of Cr and AD pathology.		[103]
Copper	9 AD patients and 5 controls (post mortem)	N/A	N.R.	AD	Presence of neuritic plaques correlated with increased Cu in senile plaque rims and cores, total senile plaques, and neuropil (measured using micro-PIXE).	Excess Cu induced mitochondrial release of ROS and activated microglia, resembling pathology in MS [99]. Cu accumulation in the spinal cord increased lipid peroxidation and may be a mechanism for ALS [99]. Cu neurotoxicity resembles AD by increasing ROS and inflammation while downregulating LRP1, resulting in reduced Aβ clearance [104]. Cu complexes with Aβ to form aggregates and impair mitochondrial function [104].	[105]
	15 WD patients and 5 controls; controls were autopsy samples collected from individuals who died in car accidents but received no brain trauma.	N/A	N.R.	WD	ICP-MS demonstrated that Cu accumulation is ubiquitous in WD neuropathology, increasing nearly 8-fold compared to controls		[106]
	17 patients with a definitive ALS diagnosis aged 47–85 years old from Oslo, Norway. 10 controls were aged 26–77 years old	N/A	N.R.	ALS	ICP-MS indicates a significantly greater CSF concentration of Cu in ALS patients compared to controls.		[94]
	N/A	*D. melanogaster* (P463, P463/*Ctr1B* RNAi, P463/*DmATP7* OE, P463/*DmATP7* RNAi)	0.25 mM copper chloride for 24 h	HD	Cu reduction rescued HD phenotypes in *D. melanogaster* (eclosion rate, shortened lifespan, impaired mobility, and degeneration of eyes and brain), whereas Cu accumulation exacerbated phenotypes. Overexpression of Cu transporters reduces protein aggregation, while silencing transporters increases protein aggregation. Feeding with Cu induced *D. melanogaster* HD phenotypes, while cotreatment with a chelator rescued this phenotype.		[107]
		Wistar rat (6-OHDA induced PD)	0.01 μg/g drinking water	PD	Increases in Cu concentrations in substantia nigra, globulus pallidus, putamen, and amygdala in 6-OHDA-treated rats compared to controls.		[108]
Uranium	17 patients with a definitive ALS diagnosis aged 47–85 years old from Oslo, Norway. 10 controls were aged 26–77 years old	N/A	N.R.	ALS	ICP-MS indicates a significantly greater CSF concentration of U in ALS patients compared to controls.	U neurotoxicity is not well defined. U induced neuroinflammation, oxidative stress, cell death, and altered neuronal signaling [109].	[94]
	N/A	*C. elegans* (NL5901, BZ555)	1 μM DU for 1–3 days	PD	*C. elegans* treated with DU exhibited increased aggregation of α-synuclein at days 3, 5, and 7. *C. elegans* exhibited shrinking of dopaminergic neurons at day 3, indicating chronic DU exposure induces neurodegeneration.		[110]
Nickel	N/A	Male Wistar rats aged 12–15 weeks	2 μL of 300 μM nickel chloride intracerebral injection	NS	Increased behaviors demonstrating anxiety and depression, as well as decreased spatial learning abilities and memory.	Multiple models show Ni accumulated in the cerebral cortex and whole brain [111]. Ni induced oxidative stress, decreased acetylcholinesterase, demyelination, and cell death in the brain [111].	[112]
	N/A	Male and female Wistar rats	0.25–0.5 mg/kg nickel chloride IP injection	NS	Increased behaviors demonstrating anxiety and depression. Worsened spatial memory.		[113]
	N/A	60 male Wistar rats aged 9 weeks old	150 μg/L nickel chloride	NS	Decreased locomotor abilities and an increase in anxiety-like behaviors. Elevated levels of Ni in the cerebrum, cerebellum, and liver. Decreased *CAT*, *GPx, GST,* and *AChE* activity. Increased *GSH* activity. Degeneration of cerebellar cortical, Purkinje, and CA3 pyramidal neurons and granule cells of dentate gyrus.		[114]
Cobalt	6 ALS patients (3 men, 3 women)	N/A	N.R.	ALS	Whole blood Co was higher in ALS patients when compared to reference values of Italian population.	Co induced neurotoxicity through DNA damage, caspase activation, increased ROS, altered mitochondrial function, apoptosis, and necrosis in neural cells [38]. Co alters neurotransmission by blocking Ca channels [38].	[91]
	17 patients with a definitive ALS diagnosis aged 47–85 years old from Oslo, Norway. 10 controls were aged 26–77 years old	N/A	N.R.	ALS	ICP-MS indicates a significantly greater CSF concentration of Co in ALS patients compared to controls.		[94]
	N/A	Male C57BL/6J mice aged 8 weeks and 48 weeks	4–16 mg/kg/day IP injection for 30 days	AD	Significant Co accumulation in blood, cerebellar cortex, and hippocampus following treatment. Increased expression of Aβ_1-42_, APP, and GSK3β. Worsened performance in cognitive and memory tasks (Morris water maze).		[115]
Zinc	9 AD patients and 5 controls (post mortem)	N/A	N.R.	AD	Presence of neuritic plaques correlated with increased Cu in senile plaque rims and cores, total senile plaques, and neuropil (measured using micro-PIXE).	Reduced clearance of Zn from glutamatergic synapses leads to amyloid oligomerization and tau protein phosphorylation in the cortex and hippocampus [116]. Zn increases aggregation rate of Aβ [116]. Zn-induced ROS is associated with lipid peroxidation in substantia nigra and mitochondrial dysfunction, associated with PD and ALS [116].	[105]
	6 ALS patients (3 men, 3 women)	N/A	N.R.	ALS	Whole blood Zn was higher in ALS patients when compared to reference values of Italian population.		[91]
	Fixed brain tissue from 8 ALS, 4 PDC, and 5 control subjects who lived in Guam.	N/A	N.R.	PD	Gray matter from PDC patients had significantly increased Zn concentrations.		[101]
	17 patients with a definitive ALS diagnosis aged 47–85 years old from Oslo, Norway. 10 controls were aged 26–77 years old	N/A	N.R.	ALS	ICP-MS indicates a significantly greater CSF concentration of Zn in ALS patients compared to controls.		[94]
	N/A	Wistar Rat (6-OHDA induced PD)	0.005 μg/g drinking water	PD	Increased Zn concentrations in substantia nigra, globulus pallidus, putamen, amygdala in 6-OHDA-treated rats compared to controls.		[108]

N.R.—Not Reported, AD—Alzheimer’s Disease, PD—Parkinson’s Disease, PDC—Parkinson’s–Dementia Complex, ALS—Amyotrophic Lateral Sclerosis, WD—Wilson’s Disease, HD—Huntington’s Disease, NS—Non-Specific Symptoms of Neurodegeneration, CSF—Cerebrospinal Fluid, ICP-MS—Inductively Coupled Plasma Mass Spectrometry, IP—Intraperitoneal, ROS—Reactive Oxygen Species, LRP1—Low-Density Lipoprotein Receptor-Related Protein 1, SOD—Superoxide Dismutase.

**Table 2 brainsci-13-00500-t002:** Summary of Senescence Markers Increased After Heavy Metal of Metalloid Exposure.

Metal/Metalloids	Model (Organ/Tissue Type)	Doses (Compound)	Duration of Exposure	Senescence Markers with Increased Expression	Method of Detection *	Refs.
Arsenic	Human Immortalized Keratinocytes, HaCaT	0.05–0.25 μM (Sodium Arsenite)	24–72 h	p16^INK4a^, p21, p53, SA-β-Gal, SASP (IL-1α, IL-6, IL-8, TGF-β1, MMP1, MMP3, EGF, VGF)	WB, qRT-PCR, ELISA, FC	[164]
	Hairless SKH1-E *p16^Luc+^* Mice (Total Body)	50 ppm (Sodium Arsenite)	12 months	p16^INK4a^	TBLI	[165]
	Human Participants (Peripheral Blood Mononuclear Cells)	30–620 μg/L	Human exposure > 10 years	p21, SA-β-Gal, SASP (IL-6, IL-8, MMP1, MMP3)	WB, ICC, EILSA, MMP1 and MMP3 Kit	[166]
	Human Articular Chondrocyte Cells, HC-a	1–5 μM (Arsenic Trioxide)	24 h	p16, p21, p53, SA-β-Gal, SASP (IL-1α, IL-1β, TGF-β, TNF-α, CCL2, PAI-1, MMP13)	WB, qRT-PCR, ICC, FC	[167]
	4-Week-Old Male Wistar Rats (Articular Cartilage)	0.05–0.5 ppm (Arsenic Trioxide)	36 weeks	p16, p21, p53, SA-β-Gal, SASP (IL-1α, IL-1β, TGF-β, TNF-α, CCL2, PAI-1, MMP13)	WB, ICC, qRT-PCR	[167]
	Mouse Skin Fibroblasts, m5S	10 ppm (Arsenic Acid)	16 h	SA-β-Gal	ICC	[168]
	Human Hepatic Stellate, LX-2	2–7.5 μM (Sodium Arsenite)	24–144 h	p21, SASP (IL-1β, IL-18, MMP1, MMP3, CXCL1)	WB, qRT-PCR, ICC	[169]
	Human Glioblastoma, U87	2–8 μM (Arsenic Trioxide)	48 h–2 weeks	p21, p53, SA-β-Gal	WB, ICC	[170]
	Human Glioblastoma, U251	2–8 μM (Arsenic Trioxide)	48 h–2 weeks	SA-β-Gal	ICC	[170]
	Human Glioma, SHG-44 cells	2–8 μM (Arsenic Trioxide)	48 h–2 weeks	SA-β-Gal	ICC	[170]
	Rat Glioma, C6	2–8 μM (Arsenic Trioxide)	48 h–2 weeks	SA-β-Gal	ICC	[170]
Lead	8-Week-Old Sprague–Dawley Rats (Hematopoietic Stem Cells)	200–600 mg/kg (Lead Acetate)	4 weeks	SA-β-Gal	ICC	[171]
	Female Sprague–Dawley Rats (Brain)	1238–2548 ppm (Lead [II] Acetate Trihydrate)	34 weeks	SASP (IL-1α, IL-1β, IL-6, IL-8, TGF-β1, TGF-β2, TGF-β3, TNF)	qRT-PCR	[172]
Cadmium	6–8-Week-Old Male Swiss Albino Mice (Spleen)	15 μM (Cadmium Chloride)	6–18 h	p21, p53, SA-β-Gal, SASP	WB	[173]
	Rat Pheochromocytoma Cells, PC12	5–20 μM	6–48 h	p16, p21, p53, SA-β-Gal, SASP (IL-1α, IL-6)	WB, ICC, qRT-PCR	[174]
	Primary Human Dermal Fibroblasts, HDF	12–24 μM (Cadmium Chloride)	4 passages	SA-β-Gal	ICC	[138]
	3–4-Week-Old Female Sprague–Dawley Rats (Bone Marrow)	50 mg/L (Cadmium Chloride)	4 weeks	SA-β-Gal	ICC	[175]
	Primary Bone Marrow Mesenchymal Stem Cells	10 μM (Cadmium Chloride)	3 h–15 days	p16, p21, SA-β-Gal, SASP (IL-1α, IL-1β, TGF-β, VEGF, CXCL-1)	WB, ICC	[175]
Chromium	Human Hepatocytes, L-02	10 nM (Potassium Dichromate)	24 h twice a week for 4 weeks	p21, p53, SA-β-Gal, SMP30, SASP (IL-1, IL-6, IL-8, G-CSF, bFGF, MMP3)	WB, ICC, qRT-PCR, ELISA	[176]
	Human Hepatocytes, L-02	4–16 μM (Potassium Dichromate)	24–48 h	p53	WB	[177]
	Prenatal Rat (Ovaries)	25 ppm (Potassium Dichromate)	5 days	p53	IHC, IF	[178]
	Human Bronchial Fibroblasts, BEAS-2B	0.5–2 μM (Potassium Dichromate)	4 weeks	SA-β-Gal	ICC	[179]
	Human Hepatocytes, L-02	10 nM (Potassium Dichromate)	3x 24 h treatments per week for 4 weeks	SMP30, SA-β-Gal, SASP (ENO1, PRDX2, COTL1, CLU)	WB, ICC, qRT-PCR, ELISA	[180]
	Human Hepatocytes, L-02	10 nm (Potassium Dichromate)	24 h treatment every other day, for 4 weeks	p21, p53, SA-β-Gal, SASP (IL-6, IL-8, GM-CSF, CXCL1, MCP-1, CLU)	WB, ICC, qRT-PCR, ELISA	[181]
	Human Hepatocytes, L-02	10 nM	3x 24 h treatments a week for 4 weeks	SA-β-Gal	ICC	[182]
*Copper*	Glioblastoma Multiforme Cells, U87-MG	100–750 μM (Copper Sulfate)	24 h	p16, p21, p53, SA-β-Gal, SASP (TGF-β1, CLU)	WB, ICC, qRT-PCR	[183]
	Human Diploid Fibroblasts, WI-38	250–1000 μM (Copper Sulfate)	24 h	p21, SA-β-Gal, SASP (TGF-β1)	WB, ICC, qRT-PCR	[184]
	Cervical Cancer Cells, HeLa	10–500 μM (Copper Sulfate)	8–16 h	p21, p53, SA-β-Gal	WB, qRT-PCR	[185]
	Human Embryonic Kidney, HEK293	10–500 μM (Copper Sulfate)	8 h	p53	WB	[185]
	Human Pulmonary Adenocarcinoma, A549	10–500 μM (Copper Sulfate)	8 h	p53	WB	[185]
	Human Bronchial Epithelial, BEAS-2B	10–500 μM (Copper Sulfate)	8 h	p53	WB	[185]
	Human Endometrial Adenocarcinoma, HEC-1-A	10–500 μM (Copper Sulfate)	8 h	p53	WB	[185]
	Human Diploid Fibroblasts, WI-38	500 μM (Copper Sulfate)	24 h	p16, p21, SA-β-Gal, SASP (TGF-β1, CTGF)	ICC, qRT-PCR	[186]
	Human Diploid Fibroblasts, WI-38	350 μM (Copper Sulfate)	24 h	p21, SA-β-Gal, SASP (TGF-β1)	WB, ICC, qRT-PCR	[187]
*Uranium*	12-Week-Old Male Sprague–Dawley Rats (Ileal Mucosa)	40 mg/L (Depleted Uranium)	3–9 months	SASP (IL- 1β, CCL2, TGF-β1, IL-10, IFNγ, COX2, PGE2)	qRT-PCR	[188]
*Nickel*	Human Diploid Fibroblasts, IMR90	300 μM–1 mM (Nickle Chloride)	48 h	p16, p21, p53	WB	[189]
*Cobalt*	Primary Human Cardiac Cells, HCM	200 μM (Cobalt Chloride)	48 h	p21, SA-β-Gal, SASP (IL-8)	WB, ICC	[190]
	Primary Human Skin Fibroblasts	0.0005:1–10,000 particle volume (μm^3^)/cell (Cobalt Chrome)	6 h–15 days	SA-β-Gal	ICC	[191]
*Silver*	Male Fischer Rats (Broncho-alveolar lavage fluid)	167 μg/m^3^–179 μg/m^3^	6 h/day for 4 consecutive days	SASP (IL-1β, MCP-1, MIP-2)	Bio-Plex Pro Assay	[192]
	Normal Human Lung Fibroblast, MRC5	0.4 μg/mL Ag^+^; 1–4 μg/mL AgNP	10 days	p21, SA-β-Gal, SASP (CSF2, CXCL3, CXCL5, CXCL8, IL- 1β, IL-6, MMP1, MMP2, MMP3, VEGFC, COX2, PGE2)	WB, ICC, IB, IF, RNA-seq	[193]
	Nine-Week-Old Male C57BL/6 Mice (Lung)	0.7 mg/m^3^ AgNP	30 min/day for 45 days	SA-β-Gal	IHC	[193]
	Human Dermal Fibroblasts, HDFa	100 μg/mL AgNP	24–144 h	p21, p53, SA-β-Gal	WB, ICC, IF	[194]
	Renal Cell Carcinoma, ACHN	100 μg/mL AgNP	24–144 h	p21, p53, SA-β-Gal	WB, ICC, IF	[194]
	Cervical Cancer Cells, HeLa	100 μg/mL AgNP	24–144 h	p21, p53, SA-β-Gal	WB, ICC, IF	[194]
*Zinc*	Vascular Smooth Muscle Cells isolated from Sprague–Dawley rat thoracic aortas (Endothelial)	10–100 μM	12 h	p21, SA-β-Gal	WB, ICC, Plate-Based Fluorescence	[195]
*Iron*	C57BL/6 mice (lung)	Intratracheal bleomycin, induced iron chelation	6 days–2 weeks	p21, p16^INK4a^	IHC	[196]
	Human Microglia Cells, SV40 Immortalized	500 μM ferric ammonium citrate	2 weeks	IL-1β, IL-6, IL-8, KV1.3	Cell Morphology, WB, ELISA	

* WB—Western Blot, ICC—Immunocytochemistry, IF—Immunofluorescence, qRT-PCR—Quantitative Real-Time Polymerase Chain Reaction, RNA-seq—RNA sequencing, ELISA—Enzyme-Linked Immunosorbent Assay, TBLI—Total Body Luciferase Imaging.

## Data Availability

Not applicable.

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
