# Peer review of "Among Gerontogens, Heavy Metals Are a Class of Their Own: A Review of the Evidence for Cellular Senescence"

_brainsci, 2023, doi:10.3390/brainsci13030500_

Round 1

Reviewer 1 Report

This is an interesting review manuscript about the role of heavy metals in cellular senescence. There are some things that work really well here, and others that need more work. Overall it is well written.

1)      There is no mention of brain aging in the abstract, and the description of aging vs brain aging is a bit confusing throughout the paper. In fact, cellular senescence is not specific to brain aging, so I’m not sure the focus on brain aging makes a lot of sense unless the authors really want to review to maintain that focus throughout the manuscript (which clearly might be needed for a publication in the journal Brain Sciences).

2)      Section 5.1 should have a more appropriate title or there should be much more about metals, cellular senescence, and neurodegenerative diseases. Perhaps a figure or table would be helpful in delineating the relationships that are known or hypothesized.

3)      Section  6 is currently not focused on brain again, but aging in general. That’s fine, but again, the focus of the manuscript needs to be more clear. In addition, some discussion of the dose at which associations are seen with cellular senescence would be useful here, making sure the dose is environmentally relevant. A table in this section might also be helpful.

4)      In general, tables and figures are what make a review paper particularly useful. They should be included here.

Minor: arsenic is a metalloid, not a metal.

Author Response

We thank the reviewer for the comments.  We have addressed comments to the best of our ability as follows:

  1. We edited the abstract to describe the focus in the context of brain aging.
  2. Title for section 5.1 was edited to be more accurate.  Table 1 was added to describe associations of heavy metals with neurodegenerative diseases.
  3. Table 2 was added to provide further details on heavy metal animal/cell culture models and associated cellular senescence observations.
  4. Two tables were added for easy reference and useful information.

Reviewer 2 Report

The review paper presents influence of heavy metals on brain and neurodegenerative processes. On the whole, the paper is well written and based on up-to-date references. Nevertheless the following points should be considered:

1.       Lack of affiliations

2.       Abstract: well-written and informative

3.       Introduction: based on up-to-date references, informative and readable. Aim of the review has been presented.

4.       Materials and methods: provided. Search strategy is adequate and consistent

5.       Section 3: Environmental heavy metal (…) should be extended. In present form, there are insufficient data.

6.       Sections 5 and 6: Please provide information which of the elements can influence on Parkinson/Alzheimer’s diseases along with mechanisms of action. Authors provided information about the diseases (section 5) but taking into account fact that the neurodegenerations are strictly related with heavy metals, the issue must be extended.

7.       Conclusions: well written

Author Response

We thank the reviewer for the comments.  We have addressed comments to the best of our ability as follows:

  1. Affiliations were added
  2. No comment to address
  3. No comment to address
  4. No comment to address
  5. Respectfully, we disagree with the reviewer.  Providing details for heavy metal environmental pollution is beyond the focus of this paper, and would require considerations for drinking water, air, soil, food sources, rural vs. urban settings, and geographical differences across countries and ecosystems.  The important focus of this section is to establish how extensive heavy metal environmental pollution is, and provide useful references for further details.  In the Tables we added, details for environmental heavy metal exposures associated with neurodegenerative diseases or brain aging were included where appropriate.
  6. We respectfully disagree with the Reviewer regarding details about the mechanisms of action.  We are discussing several heavy metals and several neurodegenerative diseases, which will involve numerous mechanisms that would require further details to describe what the mechanisms are and how metals induce dysfunction (e.g. autophagic impairment, protein aggregation, mitochondrial dysfunction, oxidative stress, DNA damage, neurotransmitter dysregulation, etc).  We believe this level of detail is beyond the scope of the paper.  Instead, we have provided further details in Tables 1 and 2 to describe mechanistic targets/observations associated with each heavy metal and references for further details.
  7. No comment to address
  8.  

Round 2

Reviewer 1 Report

Authors addressed reviewers' comments.

Author Response

No comments to address

Reviewer 2 Report

I understand that Authors did not accept my suggestions nevertheless Authors focused on the following diseases: ALS, PD, AD, WD, HD, NS ( Table 1). In my opinion, the additional table, including information about possible mechanisms (names/short basis) of metal which have impact on the diseases depelopment, should be added. Please focus on the particular mechanisms (i.e. AD : iron ions taking part in oxidation processes; PD: iron, zinc impact on a-synuclein aggregation)

Author Response

Respectfully, we disagree with the suggestion to add a table on disease mechanisms induced by metals.  Our focus is on one mechanism contributing to neurodegenerative disease: heavy metals inducing cellular senescence.  Adding a table about other mechanisms would be off target and misdirect readers.

The reviewer suggested adding details pertaining to Zn and Fe, but again this doesn't quite fit with our topic focus.  While both metals have been extensively studied in AD and PD, very few studies have considered: 1) these metals inducing cellular senescence, 2) human populations exposed to these metals and developing AD or PD, or 3) animal/cell culture studies exposing test subjects to these metals to induce cellular senescence, AD, or PD.  The vast majority of literature on this topic pertains to physiological changes in cells/tissues from patients or disease models.  However, we did add the following statement to highlight the importance of these metals (lines 255-259):

"It should be noted that two metals (iron and zinc) thoroughly discussed in PD and AD were left out of this table; these metals serve physiological roles within cells/tissues that contribute to disease process, while human exposures to these metals are not known to contribute to disease.  For reviews on the roles of iron and zinc in AD and PD, see reviews [220-224]"

We also added a section to discuss iron inducing cellular senescence, as we were able to find two additional papers (lines 421-432).  These details were also added to Table 2.